# Biotransformation of the Mycotoxin Zearalenone to its Metabolites Hydrolyzed Zearalenone (HZEN) and Decarboxylated Hydrolyzed Zearalenone (DHZEN) Diminishes its Estrogenicity In Vitro and In Vivo

**DOI:** 10.3390/toxins11080481

**Published:** 2019-08-20

**Authors:** Sebastian Fruhauf, Barbara Novak, Veronika Nagl, Matthias Hackl, Doris Hartinger, Valentina Rainer, Silvia Labudová, Gerhard Adam, Markus Aleschko, Wulf-Dieter Moll, Michaela Thamhesl, Bertrand Grenier

**Affiliations:** 1BIOMIN Research Center, Technopark 1, 3430 Tulln, Austria; 2TAmiRNA GmbH, Muthgasse 18, 1190 Vienna, Austria; 3Institute of Applied Genetics and Cell Biology (IAGZ), University of Natural Resources and Life Sciences, Vienna (BOKU), Konrad Lorenz-Straße 24, 3430 Tulln, Austria

**Keywords:** zearalenone, estrogen response element, gene expression, cell proliferation, estrogen receptor, biotransformation

## Abstract

Zearalenone (ZEN)-degrading enzymes are a promising strategy to counteract the negative effects of this mycotoxin in livestock. The reaction products of such enzymes need to be thoroughly characterized before technological application as a feed additive can be envisaged. Here, we evaluated the estrogenic activity of the metabolites hydrolyzed zearalenone (HZEN) and decarboxylated hydrolyzed zearalenone (DHZEN) formed by hydrolysis of ZEN by the zearalenone-lactonase Zhd101p. ZEN, HZEN, and DHZEN were tested in two in vitro models, the MCF-7 cell proliferation assay (0.01–500 nM) and an estrogen-sensitive yeast bioassay (1–10,000 nM). In addition, we compared the impact of dietary ZEN (4.58 mg/kg) and equimolar dietary concentrations of HZEN and DHZEN on reproductive tract morphology as well as uterine mRNA and microRNA expression in female piglets (n = 6, four weeks exposure). While ZEN increased cell proliferation and reporter gene transcription, neither HZEN nor DHZEN elicited an estrogenic response, suggesting that these metabolites are at least 50–10,000 times less estrogenic than ZEN in vitro. In piglets, HZEN and DHZEN did not increase vulva size or uterus weight. Moreover, RNA transcripts altered upon ZEN treatment (EBAG9, miR-135a-5p, miR-187-3p and miR-204-5p) were unaffected by HZEN and DHZEN. Our study shows that both metabolites exhibit markedly reduced estrogenicity in vitro and in vivo, and thus provides an important basis for further evaluation of ZEN-degrading enzymes.

## 1. Introduction

Zearalenone (ZEN), a mycotoxin produced by various *Fusarium* species, is a frequent contaminant of cereal-based foods and feeds. Although maize is regarded as the most affected commodity, ZEN can also occur in other grains, such as barley, oats, or wheat [1]. The incidence of samples tested positive for ZEN varies with factors such as region, year, commodity, or detection method [2]. A recent survey reports a ZEN occurrence of 88% in finished feed, maize, and maize silage samples collected at a global scale [3]. Although median ZEN concentrations of 0.02 mg/kg were found, individual samples reached levels of up to 11.19 mg/kg. 

ZEN acts as xenoestrogen by activation of estrogen receptors α (ERα, full agonist) and β (ERβ, partial agonist) [4]. Subsequently, ERα and ERβ translocate to the nucleus, bind to estrogen response elements (ERE, 15-bp palindrome motif) in the promotor regions of target genes and induce their transcription [5]. In vitro, this leads to a dose-dependent proliferation of estrogen-dependent cells, such as the human breast cancer cell line MCF-7, upon ZEN treatment [6]. In vivo, ZEN evokes symptoms of hyperestrogenism. In pigs, the species considered to be most sensitive to this mycotoxin, clinical signs include swelling and reddening of the vulva, metaplasia of uterus, ovarian atrophy, enlargement of the mammae in both females and males, decreased testes weight, depressed spermatogenesis and libido, reduced fertility or delivery of stillborn, and weak piglets [1,4]. Alteration of hematological and biochemical measures, gene expression and subpopulations of immunocompetent cells in tissues have been observed after ZEN exposure in piglets and prepubertal gilts, albeit with limited reproducibility between studies [7]. Recently, the effect of ZEN on the expression of microRNAs has caught scientific interest [8,9,10,11]. MicroRNAs represent a class of small non-coding RNAs, that negatively regulate gene expression. They can serve as indicators for pathological processes in organs, and possess considerable potential as biomarkers [12]. Very recently it has been shown that ZEN affects uterine expression of certain microRNAs in pigs, among them ssc-miR-424-5p, ssc-miR-450a, ssc-miR-450b-5p, ssc-miR-450c-5p, ssc-miR-503, and ssc-miR-542-3p [11]. This effect was partly attributed to estrogen receptor (ER) activation, but further molecular targets of this mycotoxin, such as G protein-coupled estrogen receptor 1 [13] or the pregnane X receptor [14], need to be considered in this context. An overview of the manifold modes of action of ZEN are provided in several comprehensive reviews [1,4,6,15,16].

Due to its frequent occurrence and impairment of animal health, effective strategies for the control of ZEN are required. Many countries worldwide introduced guidance levels for ZEN in the feed. Within the European Union, recommended guidance levels for ZEN in compound feed for pigs range from 100 μg/kg (piglets, gilts) to 250 μg/kg (sows, fattening pigs) [17]. Yet, these regulations do not account for the presence of multiple (modified) mycotoxins in feed, which might potentiate the effects of ZEN [18]. Prevention of fungal growth and toxin formation by application of good agricultural practices represents a key strategy in mycotoxin reduction management [19]. Since those preharvest measures are often insufficient to completely eliminate mycotoxins in crops, different postharvest techniques for removal of ZEN have been tested. Technological possibilities include physical and chemical methods, such as UV irradiation, gamma radiation or application of ozone and peroxide, and are able to degrade ZEN to a certain extent [20]. However, their application is hampered by factors such as the formation of unknown metabolites, nutrient losses, impairment of palatability, and cost-efficient application under field conditions [21]. Feed additives, that either adsorb or biotransform mycotoxins in the gastrointestinal tract (GIT) of animals and, thus, reduce their bioavailability and/or toxicity, seem to be more applicable in practice. Certain materials (e.g. bentonites, diatomites, zeolites, or yeasts) show remarkable adsorption efficiencies for aflatoxin B_1_, whereas binding of ZEN is comparatively low [22,23]. Modifications in the adsorbents’ surface might lead to improved binding capacities for ZEN [20], but the limited specificity of these products and the potential binding of nutrients or veterinary drugs [24] is still a major disadvantage. Hence, the biological transformation of ZEN to non- or less toxic metabolites via microorganisms or enzymes might represent a more suitable option for the feed and livestock industry.

Several fungi and bacteria originating from soil or the GIT of animals are able to transform ZEN in vitro [1,21]. One of the first studies in this field described the reduction of ZEN to α-zearalenol (α-ZEL) and to a lesser extent to β-zearalenol (β-ZEL) by rumen protozoa [25]. Since α-ZEL has a higher binding affinity to ERs than ZEN [26], this reaction cannot be considered as detoxification. Similarly, a transformation of ZEN to zearalenone-glucosides or zearalenone-sulfate, e.g. by *Thamnidium elegans*, *Mucor bainieri* or various *Rhizopus* spp. [27,28], might not be useful for further exploitation as a feed additive, as zearalenone-glucosides and zearalenone-sulfate are hydrolyzed during porcine digestion, which results in the release and subsequent systemic absorption of ZEN [29]. For many other microorganisms reported to degrade ZEN, neither the formed metabolites nor their toxicity or the responsible catabolic pathways have been elucidated in detail [1,21,30]. Two important exceptions have to be mentioned in this regard. First, *Trichosporon mycotoxinivorans* can cleave the lactone ring between C5 and C6 of the ZEN molecule, thus, producing the non-estrogenic ZOM-1 [31]. Second, *Gliocladium roseum* NRRL 1859 and *Clonostachys rosea* (synonym of *G. roseum*) IFO 7063 are capable of hydrolyzing the ester bond of the lactone ring of ZEN [32,33]. The lactonohydrolase mediating this reaction was further characterized, and the encoding gene (*zhd101*) was cloned [34]. Although the resulting metabolites had been postulated and partly confirmed before [32,33], the entire reaction mechanism was unveiled only in 2015 [30]. In this study, Vekiru et al. produced the zearalenone-lactonase Zhd101p from a codon-optimized version of the gene *zhd101*, and identified hydrolyzed zearalenone (HZEN, (E)-2,4-dihydroxy-6-(10-hydroxy-6-oxo-1-undecen-1-yl)benzoic acid) as a primary reaction product. This intermediate partly decarboxylates spontaneously to the previously described decarboxylated hydrolyzed zearalenone (DHZEN, (E)-1-(3,5-dihydroxy-phenyl)-10-hydroxy-1 -undecen-6-one, Figure 1) [30,33]. For the technological application of the zearalenone-lactonases as a feed additive, estrogenicity assessment of its metabolites HZEN and DHZEN is essential. So far, only one in vitro report on the estrogenicity of DHZEN is available, showing markedly reduced estrogenic potency in an MCF-7 proliferation assay [32]. In contrast, data on HZEN are completely lacking. 

Therefore, the aim of our study was to assess the estrogenicity of HZEN and DHZEN in vitro and in vivo, with ZEN serving as a reference. To this end, appropriate amounts of HZEN and DHZEN were produced, and subsequently tested in the MCF-7 proliferation assay as well as in a yeast bioassay. Furthermore, we evaluated the estrogenic potency of HZEN and DHZEN in prepubertal gilts based on morphological changes of the reproductive tract and alterations of mRNA and microRNA expression in uterus tissue. By generating data on the comparative estrogenicity of ZEN, HZEN, and DHZEN, our results contribute significantly to the risk assessment of this ZEN-degrading enzyme approach and its (potential) use as a feed additive.

## 2. Results

### 2.1. In Vitro Experiments

First, a proliferation assay with the human breast cancer cells MCF-7 was conducted. For that purpose, MCF-7 cells were treated with ZEN, HZEN, and DHZEN at concentrations of 0.01 to 500 nM. Cell proliferation was measured by the WST-1 assay (see Section 5.1.3). After 144 h of incubation, ZEN significantly stimulated the proliferation of human breast cancer cells at concentrations of 10, 100, and 500 nM. For comparison also ß-estradiol (E2, 10 nM) was tested, which increased cell proliferation to 226.9 ± 52% compared to the cell control. In contrast, neither HZEN nor DHZEN showed increased in cell proliferation (Figure 2). 

To confirm the lack of estrogenicity of HZEN and DHZEN in vitro, we employed a reporter gene assay with the engineered estrogen-sensitive yeast strain YZHB817. Due to the higher tolerance of the yeast cells concentrations up to 10,000 nM of ZEN, HZEN, and DHZEN were tested. Consistent with results of the MCF-7 proliferation assay, a significant estrogenic effect was detected starting at ZEN concentrations of 10 nM, whereas no estrogenicity was seen even at the highest concentrations of HZEN and DHZEN (10,000 nM; Figure 3).

### 2.2. In Vivo Experiment

The effects of dietary ZEN (4.58 mg/kg), as well as equimolar concentrations of HZEN (4.84 mg/kg) and DHZEN (4.21 mg/kg) on the reproductive tract of female prepubertal pigs, were monitored in a feeding trial. None of the treatments significantly affected the body weight of animals (Table 1). While ZEN caused a time-dependent increase of the vulva size, reaching an enlargement by a factor of 3.3 compared to the control group on day 27 [11], neither HZEN nor DHZEN influenced the vulva size during the experimental period. Similarly, the reproductive tract weight at the end of the experiment was markedly increased in the ZEN group, whereas the ZEN metabolites showed no effect on this parameter (Table 1).

In addition to investigating the effect of ZEN and its metabolites on the morphology of the reproductive tract, we also addressed potential alterations at the molecular level, by analyzing mRNA expression of estrogen-responsive genes as well as expression of ZEN responsive microRNA using qPCR.

The analysis of the porcine genomic DNA sequence for suspected estrogen-regulated genes with Dragon ERE Finder [35] resulted in the identification of fifteen potential genes with near-consensus ERE in pigs. Among them, we also found the gene EBAG9, which is one of three genes with perfect palindromic ERE identified near E2-regulated genes in the human genome. We performed mRNA expression analysis of six genes with identified EREs, and seven other genes associated with the estrogen response (see Section 5.2.2). As depicted in Table 2, ZEN exhibited only marginal effects on uterine gene expression after 28 days of exposure. The highest numerical fold change compared to the control group was observed for IL-1β (3.36-fold increase), but it lacked statistical significance due to high variability between animals. Among the genes under potential ERE activation, EBAG9 and GJA1 were significantly up-regulated upon ZEN treatment, albeit with low fold-change increase. For EBAG9, the HZEN and DHZEN groups were significantly different from ZEN group (no up-regulation). Overall, HZEN or DHZEN showed no impact on any of the other investigated genes when compared to the control group. Further statistically significant differences were only encountered between the ZEN and DHZEN groups for GAPDH and S100G, respectively. These differences can mostly be explained by high biological variability, such as seen in the control group as well.

Finally, we evaluated the impact of ZEN and its metabolites on microRNA expression in uterus. To this end, a subset of 15 microRNAs was selected for qPCR analysis based on previous data generated by Grenier et al. [11] (see Section 5.2.3 for details). Figure 4 displays the results of the hierarchical clustering analysis. While the ZEN group forms a distinct cluster, samples from the control, HZEN and DHZEN groups are heterogeneously distributed. Furthermore, individual pigs with divergent microRNA expression patterns can be deduced from this figure, e.g. pig ID 8 (Control) or ID 93 (HZEN). Analysis of data with Grubb’s test substantiated this assumption for pig ID 8, for which the critical outlier Z value of 1.8 was exceeded in ten out of 15 microRNAs. Still, since not all data were normally distributed and there was no relevant evidence to exclude this sample (e.g. impaired general health of this pig during the experiment or known issue during sample preparation), we decided to use the complete dataset for expression analysis. 

As shown in Table 3, seven of the selected microRNAs were significantly altered by the different dietary treatments. Compared to the control group, ZEN caused a significant decrease in the expression of miR-135a-5p, miR-187-3p, and miR-204-5p. In contrast, those microRNAs were unaffected in HZEN or DHZEN exposed animals. ZEN also caused numerical down-regulation of miR-129-5p and miR-149-5p, which reached statistical significance only compared to the HZEN and/or DHZEN group. None of the selected microRNAs were significantly up-regulated upon ZEN treatment, although the expression of the microRNA members of the miR-503 cluster (e.g. miR-424-5p, miR-450a-5p, miR-450b-5p, miR-450c-5p, miR-542-3p, or miR-503) showed a numerical increase of up to 4.5-fold. In the case of the miR-503 cluster, the only statistically significant difference was detected for miR-450c-5p between the ZEN and DHZEN group. In Figure 5, Ct normalized values of four miRNAs are depicted to provide a better overview of the inter-individual variance of microRNA expression, and thus the potential inter-individual difference in response to the administered substances. Similar to the mRNA analysis, neither HZEN nor DHZEN caused significant alterations of any investigated microRNAs compared to the control group. 

## 3. Discussion

Due to the limitations of traditional physical and chemical strategies for the detoxification of mycotoxins, biodegradation of mycotoxins by microorganisms and/or enzymes is an attractive approach to reduce the negative effects of mycotoxins. Several microorganisms capable of detoxification of ZEN and other mycotoxins have been previously reported [1,20]. However, for application as a feed additive, feed safety issues have to be sufficiently addressed. From a technological point of view, preparation of microorganisms with good stability during storage and high activity in the gastrointestinal tract is difficult. In comparison, enzymatic degradation using highly specific enzymes should be more effective. An example is the zearalenone-lactonase Zhd101p, which degrades ZEN to HZEN as the primary reaction product. HZEN is unstable and partially decarboxylates to DHZEN. To evaluate the technological potential of the zearalenone-lactonase as a feed additive, the present study aimed to investigate the estrogenic activity of its reaction products HZEN and DHZEN in vitro and in vivo. 

ER-positive MCF-7 cells are widely used in endocrine research, e.g. to determine the estrogenic activity of substances (proliferation assay, also known as E-screen) or to study potential remedies for estrogen-related diseases, such as breast cancer. Various studies have demonstrated a dose-dependent pro-proliferative effect of ZEN in this cell line (e.g. [32,36,37,38]). Likewise, ZEN increased cell proliferation of MCF-7 cells in our experiment, which became statistically significant at a concentration of 10 nM (201.4 ± 53.0% compared to the control set to 100%). At higher ZEN concentrations, a saturation in cell proliferation at approximately 220% was observed, which is in line with previous studies, which report a plateau at around 170% [38] to 220% [32]. In contrast, neither HZEN nor DHZEN affected cell proliferation in the tested concentration range, indicating that these metabolites are at least 50 × less estrogenic than ZEN (500 nM HZEN showed no effect, whereas 10 nM ZEN showed a significant effect). Higher concentrations were not tested, since ZEN starts to impair cell viability in breast cancer cell lines at 1000–25,000 nM [36,37,39]. So far, only one report is available on the estrogenicity of one of those ZEN metabolites: Kakeya et al. [32] showed that DHZEN has no impact on cell proliferation of MCF-7 cells up to a concentration of 100 nM. Thus, we could not only confirm these results but extend them to HZEN, which was identified as the primary reaction product of the zearalenone-lactonase only a few years ago [30]. 

The MCF-7 proliferation assay only provides information on whether a substance evokes the cellular response known to be induced by estrogens, whereas definite conclusions on the underlying mechanisms cannot be drawn [40]. Hence, we used a yeast bioassay to investigate specifically the effect of ZEN, HZEN, and DHZEN on ER activation and subsequent transcription of the reporter gene *GAL7-lacZ*. Moreover, this yeast bioassay provides several additional advantages, e.g. much easier to perform, no cell culture equipment is needed, the concentration range of test substances is broader, and it has a comparably low turnover time (incubation with test substances takes 18 h as compared to 144 h for MCF-7 proliferation assay). In the present study, ZEN significantly increased the lacZ activity at 10 nM (124.1 ± 13.8% compared to the control set to 100%). In the MCF-7 proliferation assay, estrogenicity of ZEN was detected at the same concentration. It has been reported that yeast bioassays can be less sensitive than the MCF-7 proliferation assay [40], but this was not observed in our study. In line with results of the MCF-7 proliferation assay, HZEN and DHZEN did not activate the *GAL7-lacZ* reporter gene in the yeast bioassay, even at the highest concentration tested (10,000 nM). Given the tested concentration range, HZEN and DHZEN can be considered as at least 1000 times less estrogenic than ZEN in this assay.

Collectively, our in vitro data showed that neither HZEN nor DHZEN have any measurable effect on MCF-7 cell proliferation or yeast reporter gene transcription in the tested concentration range, indicating a low or even absent intrinsic estrogenic activity of those metabolites. The highest levels used (500/10,000 nM) correspond to 168/3364 ng/mL for HZEN, 146/2924 ng/mL for DHZEN and 159/3184 ng/mL for ZEN. To compare with in vivo exposure, these concentrations were checked against levels of ZEN found in biological samples (ZEN was used as a reference, given no toxicokinetic data are available for HZEN or DZHEN). For instance, one of the highest tissue levels of ZEN measured in pigs was reported by Doell et al. [41], who found on average 5.3 ng/g ZEN in the liver after dietary administration of 0.42 mg/kg ZEN. This tissue concentration is approximately by a factor 30/600 lower than the highest toxin level used in our in vitro experiments. In addition, the aforementioned tissue concentration represents a sum value of free and glucuronidated ZEN. The glucuronidation rate in liver accounts for at least 62% [42] and ZEN-glucuronides lack substantial estrogenic activity [43], further strengthening the assumption that HZEN and DHZEN do not exhibit estrogenic activity at practically relevant concentrations in vivo. However, as emphasized by Andersen et al. [40], the estrogenic activity of a substance in an intact organism cannot be directly deduced from in vitro results. Important factors that need to be considered in this regard comprise the substance’s absorption, distribution, metabolism and excretion (ADME), its ability to enter target cells and the concentration of endogenous estrogens. To underline this, differences in the intrinsic activity between ZEN, α-ZEL, and β-ZEL are well documented [4], whereas variations in plasma protein binding among these substances and among species were unraveled only recently [44]. Hence, to indirectly account for all these factors, we evaluated the estrogenic activity of HZEN and DHZEN also in an in vivo experiment in pigs. 

Prepubertal female pigs were exposed to 4.58 mg/kg ZEN and equimolar concentrations of HZEN and DHZEN, respectively, in order to identify potential adverse health effects of ZEN metabolites at relatively high dietary levels. This concentration clearly exceeds the EU recommendations for ZEN in feed [17], but might still be encountered under unfavorable conditions, where up to 11.19 mg/kg ZEN was found in an individual feed sample [3]. In our experiment, the effect of ZEN treatment on vulva size enlargement and reproductive tract weight increase was prominent. This was expected, as already exposure to 1.3 mg/kg ZEN for 24 d was shown to significantly affect reproductive tract morphology [45]. In contrast to ZEN, neither HZEN nor DHZEN caused alterations in any of the clinical parameters assessed, suggesting that both metabolites lack relevant estrogenic activity in vivo. To substantiate this assumption, we next focused on potential changes on the molecular level. Since previously reported data of this animal experiment showed that ZEN had no influence on the plasma metabolome [46], we examined the impact of different treatments on the target organ uterus using a transcriptomics approach.

The classical mechanism, in which ligand-bound ERs interact directly with estrogen response elements (EREs) to activate the transcription of genes, is called the genomic response. Although less extensively studied than E2, ZEN has been shown to induce an estrogenic response via this mechanism [47], and therefore, we first investigated the expression of several genes with identified or potential EREs in their DNA sequences. As in the human homolog, a perfect ERE palindrome was found in the porcine gene sequence of EBAG9. The ERE of the human homolog is recognized by ERs with high specificity in vitro [48]. Although a significant increase was observed for the transcription of this gene after ZEN ingestion, the effect seems small (1.5-fold change) given the high concentration of ZEN used. In contrast, EBAG9 expression in HZEN and DHZEN groups were not different from the control group. Similar to EBAG9, little effect was seen on the expression of GJA1, for which a potential ERE was identified. GJA1 has been already reported as highly regulated by E2 in rat endometrium [49]. In addition, the authors mentioned that the expression of the C3 complement and the ODC genes were augmented in the uterus of rodents exposed to endocrine disruptors. Based on our results, which show no significant impact of ZEN on the uterine expression of C3 and ODC, we cannot confirm these findings in pigs for this xenoestrogen. Interestingly, the uterine transcription of GAPDH, which is frequently used for qPCR normalization, was affected by ZEN treatment (however, GAPDH transcription in the liver and jejunum was not affected, data not shown). Thus, GAPDH was excluded from the list of housekeeping genes we used. This effect was already described in 1998 by Zou and Ing [50] in the endometrium of ewes exposed to E2, and it emphasizes the need to select suitable reference genes for each analysis of mRNA expression. Although our initial hypothesis on the activation of genes carrying ERE motifs upon ZEN exposure is not conclusive, activation of GREB1 containing ERE in its promoter was very recently demonstrated on MCF-7 cells treated with ZEN [51]. In this study, authors used chromatin immunoprecipitation to show that ZEN induced the recruitment of ERα DNA-binding at chromatin sites. It is plausible that the genes we selected are carrying elements that may not represent binding sites in vivo, possibly because of chromatin accessibility [48].

In line with that, we also extended our analysis to genes otherwise associated with the response to estrogens. Several studies showed that the gene S100G, better known to encode the protein Calbindin-D9k (CaBP-9k), is a potent biomarker for screening estrogen-like environmental chemicals [49,52]. In pigs, E2 treatment induced an increase in CaBP-9k mRNA levels, while exposure to progesterone caused a decline in the expression [53]. It seems that CaBP-9k expression is also fluctuating depending on the serum estrogen levels [52]. In our experiment, none of the treatments induced significant alterations in uterine CaBP-9k expression compared to the control group. Yet, the opposite effects seen between the piglets fed ZEN and DHZEN deserve further investigations. As briefly discussed in Grenier et al. [11], the lack of effects observed on the ERα is in agreement with Oliver et al. [54], who analyzed the gene and protein expression of ERs in the uterus of piglets exposed to 1.5 mg/kg of ZEN for four weeks. However, Oliver et al. reported a two-fold increase in the mRNA level of ERβ in those animals, which could not be confirmed in the present experiment. 

Another transcriptomics approach we implemented in the present study is the targeted analysis of selected microRNAs. Unlike mRNA analysis focusing on the direct activation of gene transcription, this approach gives information on post-transcriptional gene regulation. Indeed, changes in microRNA levels (both in blood and tissue) have been reported for certain estrogen-associated diseases in humans, such as breast cancer [55]. In vitro experiments, e.g. in MCF-7 cells, underlined the role of microRNAs not only in estrogen signaling, but also in mediating the effects of different endocrine disruptors [55]. Since 2015, several reports on the impact of ZEN on microRNA expression have become available, addressing changes in microRNAs levels in the porcine liver and intestine [9], porcine pituitary gland and its consequences for gonadotropin regulation [10], murine Leydig cell line TM3 [8], as well as porcine uterus and serum [11]. Based on results obtained in the latter, we selected 15 microRNAs and evaluated their response to ZEN, HZEN, and DHZEN treatment by qPCR. 

As shown in Table 3, seven of the selected microRNAs were significantly altered by the different dietary treatments. Compared to the control group, ZEN caused a significant decrease in the expression of miR-135a-5p, miR-187-3p, and 204-5p. In contrast, those microRNAs were unaffected in HZEN or DHZEN exposed animals. Previous literature studies identified binding sites for ERα in the microRNA gene hsa-miR-135a2, and subsequent in vitro experiments showed up-regulation of miR-135a-5p (previous name: miR-135a) in E2-treated MCF-7 and ZR-75-1 cells [56,57]. Hence, our in vivo findings on decreased miR-135a-5p levels after ZEN exposure were surprising. Yet, direct comparisons of in vivo and in vitro results on the microRNA response after exposure to ER agonists should be done with caution. First, most of the in vitro studies use cancer cell lines and among those studies, conflicting results on the microRNA response were obtained due to differences in methodology and potential time-dependent effects [55]. Also, the vast majority of in vitro studies only investigate the role of single microRNAs, which does not reflect the interplay of various microRNAs or potential regulatory feedback-loops. As a consequence, it remains challenging to determine the specific role of an individual microRNA in the complex estrogen response (which is not solely limited to ERα and ERβ activation) and related diseases. For example, miR-135a-5p was reported to promote, as well as to decrease, breast cancer cell migration [58,59]. Likewise, miR-204-5p (previous name: miR-204) has been addressed both as oncogene and tumor suppressor, and is e.g. aberrantly expressed in endometrial carcinoma [60]. In ovariectomized mice, uterine expression of miR-204 was decreased after E2 treatment [61]. This effect was counteracted by pre-treatment with ER antagonists, which indicates that microRNA deregulation was mediated via the ER pathway. 

Although miR-135a-5p, miR-187-3p, and 204-5p have been associated with estrogen signaling or related diseases, they do not seem to represent the key players in estrogen signaling [5,55]. In this regard, members of the miR-503 cluster (a set of two or more microRNAs transcribed from physically adjacent microRNA genes) were suggested to have a more prominent role, with miR-503 itself being addressed as “candidate master regulator of the estrogen response” [62]. Although a numerical increase was seen in the present study for the microRNAs belonging to this cluster (e.g. miR-424-5p, miR-450a-5p, miR-450b-5p, miR-450c-5p, miR-542-3p, or miR-503), no significant (*p* < 0.05) effect was seen in the uterus of pigs exposed to ZEN, and only some of these microRNAs showed a trend (*p* < 0.1) towards increased expression. This can be explained by a high inter-individual variance in expression level, with certain pigs being very responsive (up to 8-12-fold increase). Although this increases the variability within the group, the same consistent effect was seen across these six microRNAs (similar average in expression around 3.5-4.0-fold increase). These results are yet not as pronounced as seen for these microRNAs when using an untargeted RNA sequencing approach [11]. Since samples were derived from the same feeding trial, these differences in effect intensity and significance might be due to different approaches used for microRNA quantification (qPCR and RNA sequencing). Overall, none of the investigated microRNAs were significantly altered upon HZEN or DHZEN treatment compared to the control group. A numerical increase of certain microRNAs (e.g. miR-450c-5p or miR-542-3p) was observed after HZEN treatment, which most likely is irrelevant at practically relevant metabolite concentrations. Still, follow-up studies should address this aspect, as well as the time-dependent effects of ZEN on the microRNA expression. Already now, generated results can serve for prediction and subsequent experimental verification of new mRNA targets of ZEN. 

Key steps in the development of mycotoxin-degrading enzymes are the identification of enzymes with a certain degradation potential, the characterization of the produced metabolites as well as the elucidation of their toxicities [19]. All those conditions have now been met for the zearalenone-lactonase Zhd101p. For the technological application of this enzyme as a feed additive, further challenges must be met, above all the fast and efficient degradation of ZEN in vivo. In addition, the enzyme’s safety (for workers, target species, consumers, and environment) and its storage stability must be demonstrated [63]. To the best of our knowledge, no ZEN-degrading enzyme has fulfilled all those criteria so far and received EU authorization [64]. Hence, further research efforts are needed to develop a safe and effective ZEN-degrading enzyme to be used as a feed additive. Furthermore, ZEN poses not only a risk for animal health, but is also of concern for humans [20]. In future, the enzyme technology might be evaluated for its application in human nutrition, e.g. during food processing [65]. Importantly, such developments should never indirectly promote the production of unsafe raw materials, but rather be considered complementary to measures for the reduction of mycotoxin formation pre- and postharvest [66]. 

## 4. Conclusions 

HZEN and DHZEN exhibited markedly diminished estrogenic activity compared to ZEN in both the MCF-7 proliferation assay and the yeast bioassay. In vivo, these metabolites did not cause morphological changes in the reproductive tract in prepubertal gilts. Conclusions on their molecular effects are hampered by the limited impact observed for ZEN, especially on mRNA level. Still, for transcripts altered upon ZEN treatment (EBAG9, miR-135a-5p, miR-187-3p, and miR-204-5p), neither HZEN nor DHZEN showed an effect. Hence, our data indicate that cleavage of ZEN by the zearalenone-lactonase Zhd101p reduces its estrogenicity in vitro and in vivo. Our study represents an important step in the safety evaluation of ZEN-hydrolyzing enzymes.

## 5. Materials and Methods 

### 5.1. In Vitro Experiments

#### 5.1.1. Chemicals, Reagents and Materials 

Solid ZEN was obtained from Fermentek LTD (Israel, purity 99.2%), while β-estradiol (E2) was purchased from Sigma-Aldrich (St. Louis, MO, USA). Further chemicals and reagents used for HZEN and DHZEN production comprise acetonitrile (ACN, chromasolv for HPLC, ≥ 99.9%, Sigma-Aldrich, St. Louis, MO, USA), hydrochloric acid 37% (puriss. p.a) and tris base (both Sigma-Aldrich, St. Louis, MO, USA). 

For MCF-7 experiments, charcoal-stripped fetal bovine serum and dimethylsulfoxid (DMSO, both Sigma-Aldrich, St. Louis, MO, USA), fetal bovine serum (FBS) and insulin-transferrin-selenium-A supplement (ITS, both Thermo Fisher Scientific, Waltham, MA, USA), L-glutamine (Sigma-Aldrich, St. Louis, MO, USA) as well as RPMI 1640 with phenol red and RPMI 1640 without phenol red (both Sigma-Aldrich, St. Louis, MO, USA) were used. Cell proliferation reagent WST-1 was obtained from Roche (Mannheim, Germany).

For the yeast bioassay, chemicals and reagents include α-D-(+)-Glucose (water free, 96%) and 2-mercaptoethanol (both Sigma-Aldrich, St. Louis, MO, USA), 2-nitrophenyl-B-D-galactopyranosid (ONPG) (Roth, Karlsruhe, Germany), amino acid drop out mix (adenine, arginine-hydrochloride, aspartic acid, glutamic acid monosodium salt monohydrate, histidine monohydrochloride, isoleucine, leucine, lysine, methionine, phenylalanine, serine, threonine, tyrosine, uracil, valine, all amino acids in L-form and purchased from Sigma-Aldrich, St. Louis, MO, USA), magnesium sulfate, sodium carbonate anhydrous, sodium dihydrogen phosphate monohydrate, sodium phosphate (dibasic) and potassium chloride (all Sigma-Aldrich, St. Louis, MO, USA), Y-PER Yeast Protein Extraction Reagent (Sigma Aldrich, St. Louis, MO, USA), and yeast nitrogen base (Invitrogen, Carlsbad, CA, USA). 

#### 5.1.2. Production of HZEN and DHZEN

Production and purification of HZEN and DHZEN for in vitro and in vivo experiments were done according to Vekiru et al. [30] with slight modifications. Briefly, multiple batches of 50 mg ZEN each were weighed in glass bottles and dissolved in 50 mL of ACN. Afterwards, 1 L of Tris-HCl buffer (pH value 7.5, 30 °C) and 1.2 mg of a codon-optimized version of the ZEN-lactonase (Zhd101p) of *Gliocladium roseum* were added. For the production of HZEN, solutions were incubated on a magnetic stirrer for 38 h at 30 °C. For DHZEN production half of the HZEN solution was used. Decarboxylation of HZEN was achieved by adding HCl to the HZEN solution to yield a final pH value of 3, and further incubation for 10 h at 50 °C. Both solutions, HZEN and DHZEN, were concentrated by solid phase extraction and purified by preparative HPLC as described by Vekiru et al. [30]. Purity (HZEN: 95%, no residual ZEN detected, DHZEN: 98%, no residual ZEN or HZEN detected) and quantity of ZEN metabolites were assessed by LC-UV-MS/MS as described by Hahn et al. [67].

#### 5.1.3. MCF-7 Proliferation Assay

The human breast adenocarcinoma cell line MCF-7 was purchased from the German Collection of Microorganisms and Cell Cultures (DSMZ, No. ACC 115, Braunschweig, Germany) and grown in RPMI 1640 with phenol red supplemented with 10% FBS, 1% insulin-transferrin-selenium-A supplement, 1% L-glutamine, and 1% sodium pyruvate at 37 °C and 5% CO_2._ Three days prior to the experiment, cultivation medium was replaced by the hormone-free medium (RPMI 1640 without phenol red, supplemented with 10% charcoal-stripped FBS, 1% insulin-transferrin-selenium-A supplement, 1% L-glutamine, and 1% sodium pyruvate). Approximately 5000 cells/well were seeded in hormone-free medium (200 µL/well) into a 96-well microplate and cultivated at 37 °C and 5% CO_2_ for 48 h. 

In total, MCF-7 cells were subjected to eight different treatments. In five independent experiments, the effects of ZEN and its metabolites on cell proliferation were tested in six concentrations (0.01, 0.1, 1, 10, 100, and 500 nM) and compared to a negative (solvent) and positive (10 nM E2) control group. To this end, stock solutions of ZEN, HZEN, and DHZEN (in DMSO) were diluted with hormone-free medium to the respective concentrations. The DMSO concentration was kept constant at 0.05% in all dilutions, and thus was also used as a negative control. 

After 144 h of incubation with different substances, the cell proliferation assay WST-1 (4-[3-(4-iodophenyl)-2-(4-nitrophenyl)-2H-5-tetrazolio]-1,3-benzene disulfonate) was performed according to the manufacturer’s instructions. Briefly, supernatants were discarded and cells were incubated with a 10% WST-1 solution in the hormone-free medium at 37 °C and 5% CO_2_ for a maximum of 4 h. Absorbance (A_450_) was measured by a microplate reader (Synergy HT, Biotek, Winooski, VT, USA). The development of formazan dye correlates to the number of metabolically active cells in the culture.

#### 5.1.4. Yeast Bioassay 

The yeast strain YZHB817 is a derivative of the yeast two-hybrid strain PJ69-4A [68] in which the genes encoding the ABC transporter proteins Pdr5 and Snq2 were deleted to reduce efflux and increase net ZEN uptake [69]. The *GAL7*-promoter dependent *lacZ* reporter gene expression is activated by a hybrid protein consisting of amino-acids 1-848 of the yeast Gal4p (activator of galactose utilization genes, DNA binding domain) and amino-acids 282-595 of the human ERα providing the hormone dependent activation domain. This chimeric protein is expressed behind the constitutive *ADH1* promoter on the centromeric plasmid (*CEN4 TRP1*) pTK103 [70].

A single colony of YZHB817 grown on an SC-TRP agar plate was used to inoculate 25 mL SC-TRP medium (prepared according to Sherman [71]) and incubated overnight at 28 °C, 200 rpm. Dilutions of ZEN, HZEN, and DHZEN were prepared in SC-TRP medium resulting in the following concentration levels: 0, 1, 10, 100, 500, 1000, 5000, 10,000 nM. The DMSO concentration was kept constant at 1% in all dilutions. For substance testing, the yeast was diluted to an OD of 0.1 measured at 600 nm and 100 µL aliquots were transferred into U-bottom 96-well plates. After centrifugation (15 min, 2250 rcf, 12 °C) the supernatant was discarded. Next, 100 µL aliquots of the diluted samples were transferred into U-bottom 96-well plates containing the YZHB817 cells. All samples were tested in quadruplicate per plate, and the experiment was repeated four times. The plates were covered with a sterile BREATHseal™ (Greiner Bio-One, Kremsmünster, Austria) and incubated at 28 °C, 70% RH, 800 rpm for 18 h.

For detection of ß-galactosidase expression, the reaction reagent was prepared as follows: Y-Per, Z-buffer 10 × (60 mM Na_2_HPO_4_ × 7 H_2_O, 40 mM NaH_2_PO_4_ × H_2_O, 10 mM KCl, 1 mM MgSO_4_, 50 mM 2-mercaptoethanol, pH 7.0) and ONPG-stock (8 mg/mL 2-Nitrophenyl β-D-galactopyranoside) were mixed at a ratio of 0.45/0.275/0.275. Subsequently, 35 µL of this reagent was added to each well and mixed. The plates were incubated for 10 min at 37 °C until the samples turned slightly yellow. The reaction was stopped by the addition of 25 µL of 2 M Na_2_CO_3_ per well. The plates were centrifuged for 15 min at 2250 rcf, 12 °C and 60 µL of the supernatants were transferred into flat bottom 96-well plates. Finally, the absorbance was measured at 420 nm.

### 5.2. In Vivo Experiment 

#### 5.2.1. Animals and Study Design

The animal experiment was conducted at the Center for Animal Nutrition (Waxenecker KEG, Austria) and approved by the Institutional Ethical Committee and the Lower Austrian Region Government, Group Agriculture and Forestry, Department of Agricultural Law (LF1-TVG-39/017-2015). All experimental procedures were carried out in accordance with the European Guidelines for the Care and Use of Animals for Research Purposes [72] and the Austrian Animal Experimentation Act 2012. The approval dates for the animal experiment were 30.6.2015 (Institutional Ethical Committee) and 29.07.2015 (Lower Austrian Region Government, Group Agriculture and Forestry, Department of Agricultural Law (LF1-TVG-39/017-2015).

As described in Grenier et al. [11], female weaned piglets (sow: Landrace × Large White, boar: Pietrain, 30 ± 2 days old) were obtained from a local producer and allowed to acclimatize for seven days. Piglets (n = 24) were housed on slatted floor pens under controlled environmental conditions, and their general health status was monitored daily. To compare the effects of ZEN and equimolar concentrations of its metabolites, piglets were exposed to one of the four experimental diets (n = 6): i) uncontaminated feed (Control), ii) feed containing 4.58 mg/kg ZEN (ZEN), iii) feed containing 4.84 mg/kg HZEN (HZEN), or iv) feed containing 4.21 mg/kg DHZEN (DHZEN). Experimental diets and water were provided *ad libitium* for 28 days. Individual bodyweight, vulva length and vulva width of piglets were monitored in regular intervals during the experimental period. As presented in detail in Grenier et al. [11], animals were euthanized at day 28, the reproductive tract was dissected and weighed, and uterus samples were collected for qPCR analysis (mRNA, microRNA). 

For homogenous distribution of HZEN and DHZEN within the feed, lyophilizates (Section 5.1.2) were first mixed with maltodextrin (proportion 1:8 and 1:6, respectively). Subsequently, these premixes were added to the basal feed at an inclusion rate of 0.9%. Concentrations of final diets were confirmed by LC-UV-MS/MS. The absence of any natural and significant contaminations with mycotoxins in the basal feed, as well as the procedure for artificial ZEN contamination of the diet, is described in Grenier et al. [11].

#### 5.2.2. qPCR Analysis of mRNAs From Genes with Potential ERE or Associated with Estrogen Response

E2-liganded ERs bind with the highest affinity to EREs (15-bp palindromes) composed of PuGGTCA motifs separated by three variable bp [73]. To a lesser extent than the extensive work done by Bourdeau et al. [48] in human and mouse, we tried to identify near-consensus ERE sequences in pig based on human E2-responsive genes. Sixty genes were pulled out from the list of Bourdeau et al. [48] on the E2-upregulated genes (experimentally validated in humans), and the respective sequences of these genes in pigs were obtained (from Ensembl database, https://www.ensembl.org/index.html). Subsequently, each genomic DNA sequence was analyzed with the program Dragon ERE Finder version 2 [35] in order to identify potential EREs. In addition to this hypothesis of ZEN activating genes carrying ERE on their sequences, we also selected a few genes (Table 4) showing growing evidence in the literature that they are responsive to estrogenic compounds, such as the gene S100G (also known as CaBP-9k). We did not find any potential ERE motifs on the DNA sequences of those genes. The expression of other genes such as ERα, ERβ, and the two pro-inflammatory cytokines IL-1β and IL-6 was also evaluated via RT-qPCR.

Beforehand, isolation of total RNA was done approximately from 30 mg of uterus tissue, disrupted via bead-beating and RNA was extracted with the miRNeasy Mini Kit (Qiagen, Hilden, Germany) according to the manufacturer’s recommendations. The total RNA was used for first-strand cDNA synthesis using Maxima H Minus First Strand cDNA synthesis kit including dsDNase (Thermo Fisher Scientific, Waltham, MA, USA) according to standard procedures. A few primers were used from literature but most of them were designed with the software Primer3 ([74], https://primer3plus.com/primer3web/primer3web_input.htm), and pre-experimentally validated. In addition, due to a differential effect of the ZEN treatment on the expression of housekeeping genes, we used the software GeNorm [75] to select among six candidates (ActB, GAPDH, HPRT1, RPL4, RPL32, and TBP) the most stable ones for data normalization (Table 4). All qRT-PCR reactions were conducted on the Mastercycler ep Realplex (Eppendorf, Hamburg, Germany) using SYBR green chemistry (Kapa SYBR Fast Universal, Sigma-Aldrich, St. Louis, MO, USA). The thermal cycle conditions were as follows: 1 cycle of pre-incubation at 95 °C for 3 min, 40 cycles of amplification (95 °C for 10 sec, 60 °C for 20 sec, and 72 °C for 20 sec), and melting curve program included at the end of the run. Relative gene expression was calculated using the 2^ΔΔCt^ method with the geometric mean of the Cts from the four housekeeping genes serving for normalization (subtracting the Ct-values of individual target genes from the Ct-value of the housekeeping genes of the same sample (ΔCt-values). Next, the mean ΔCt for each experimental group and target gene was calculated, and subsequently used for statistical evaluation and expressing the fold change (= 2^ΔΔCt^ value).

#### 5.2.3. qPCR Analysis of microRNAs 

To compare the effects of ZEN and its metabolites on uterine microRNA expression, microRNAs previously described to be altered by ZEN exposure [11] were chosen for targeted qPCR analysis. In total, 15 microRNAs were selected based on expression patterns (magnitude of log2 fold changes upon ZEN treatment, average tags per million count) observed after sequencing of uterus tissue. The panel included eight microRNAs up-regulated upon ZEN exposure (ssc-miR-1, ssc-miR-143-3p, ssc-miR-424-5p, ssc-miR-450a, ssc-miR-450b-5p, ssc-miR-450c-5p, ssc-miR-503, and ssc-miR-542-3p) and seven down-regulated microRNAs (ssc-miR-129a, ssc-miR-135, ssc-miR-149, ssc-miR-181c, ssc-miR-187, ssc-miR-204-5p and ssc-miR-206). 

The extraction of RNA was done as described in the above section for mRNA. Synthesis of cDNA and PCR amplification was performed as described in Grenier et al. [11]. The method to calculate the fold change is the same as for the mRNA, the 2^ΔΔCt^ method. Normalization of the Ct values was performed with the reference U6 snRNA, subtracting the Ct-values of individual microRNAs from the Ct-value of U6 snRNA of the same sample (ΔCt-values). 

### 5.3. Statistics

Data from the MCF-7 proliferation assay and the yeast bioassay were expressed relative to the negative control, which was set to 100%. Statistical analysis was performed with SigmaPlot (Version 12.5, from Systat Software, Inc., San Jose, CA, USA). Since data were not normally distributed (Shapiro-Wilk test, *p* < 0.05), a nonparametric Kruskal-Wallis test (Dunn’s multiple comparison post-hoc test) was conducted to assess the effects between different concentrations to the cell control.

Data from the feeding trial (body weight, reproductive tract weigh, vulva size, mRNA, and microRNA expression) were analyzed with IBM SPSS Statistics (Version 22, Armonk, NY, USA). If data were normally distributed (Shapiro-Wilk test, *p* > 0.05), a one-way ANOVA was conducted to assess the effects of treatment diets on dependent variables. When main effects were significant, differences between means were examined via Tukey post-hoc analysis (assumption of homogeneity of variances met, Levene’s test, *p* > 0.05) or Games-Howell test (no homogeneity of variance). Data that were not normally distributed were analyzed with the non-parametric Kruskal-Wallis test (Dunn-Bonferroni post-hoc test). Differences between means were considered significant at *p* < 0.05. GraphPad Prism version 8 for Windows (GraphPad Software, La Jolla California USA) was used for running the Grubbs’ outlier test and generating figures. 

## Figures and Tables

**Figure 1 toxins-11-00481-f001:**
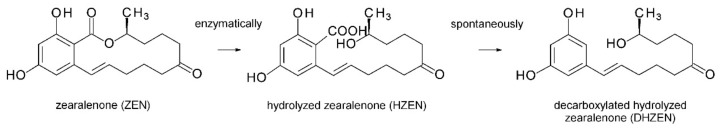
Structures of zearalenone (ZEN) and its metabolites hydrolyzed zearalenone (HZEN) and decarboxylated hydrolyzed zearalenone (DHZEN) produced by the zearalenone-lactonase Zhd101p (modified based on [30]).

**Figure 2 toxins-11-00481-f002:**
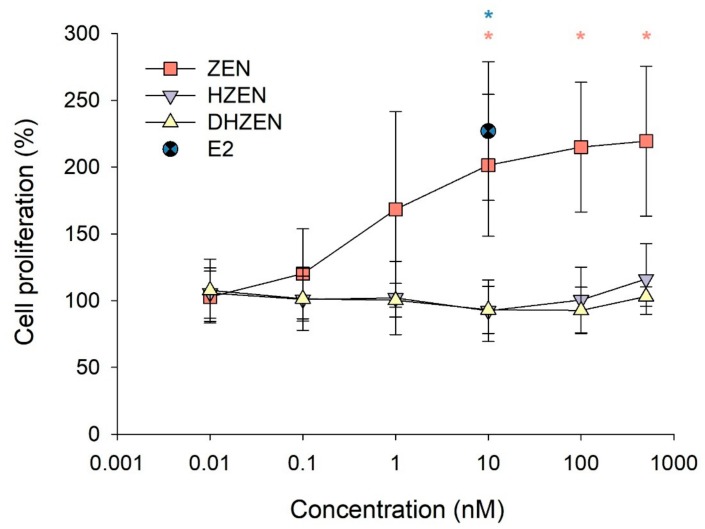
MCF-7 cell proliferation (%) after treatment with 0.01, 0.1, 1, 10, 100, and 500 nM ZEN (positive control), HZEN or DHZEN for 144 h. Cell control (= 0.05% DMSO) was set to 100%. 17ß-estradiol (E2, 10 nM) served as additional positive control. Each result represents the mean ± SD of five independent experiments (with three technical replicates per experiment). Significant differences are indicated with * (*p* < 0.05). The blue * only concerns the comparison between Control and E2.

**Figure 3 toxins-11-00481-f003:**
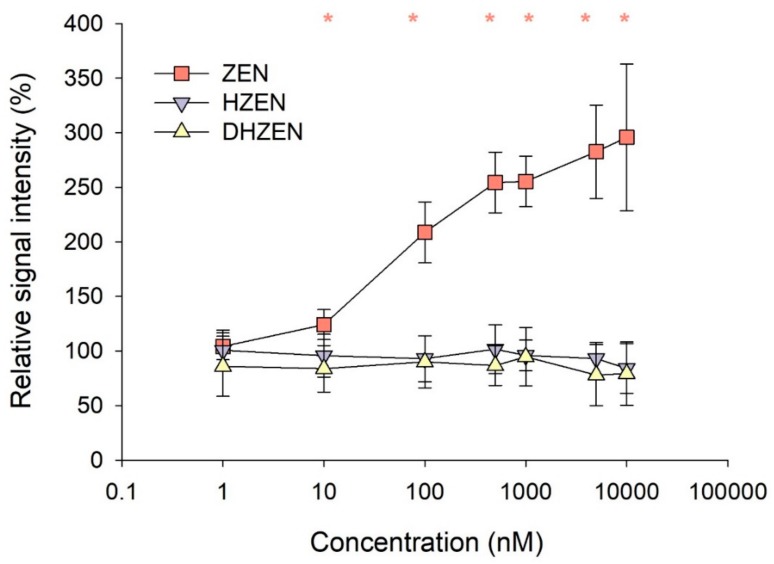
Estrogen dependent ß-galactosidase activity in yeast bioassay after treatment with 1, 10, 100, 500, 1000, 5000 and 10,000 nM ZEN, HZEN, or DHZEN for 18 h. Signal intensity of cell control (= 1% DMSO) was set to 100%. Each result represents the mean ± SD of four independent experiments (with four technical replicates per experiment). Significant differences are indicated with * (*p* < 0.05).

**Figure 4 toxins-11-00481-f004:**
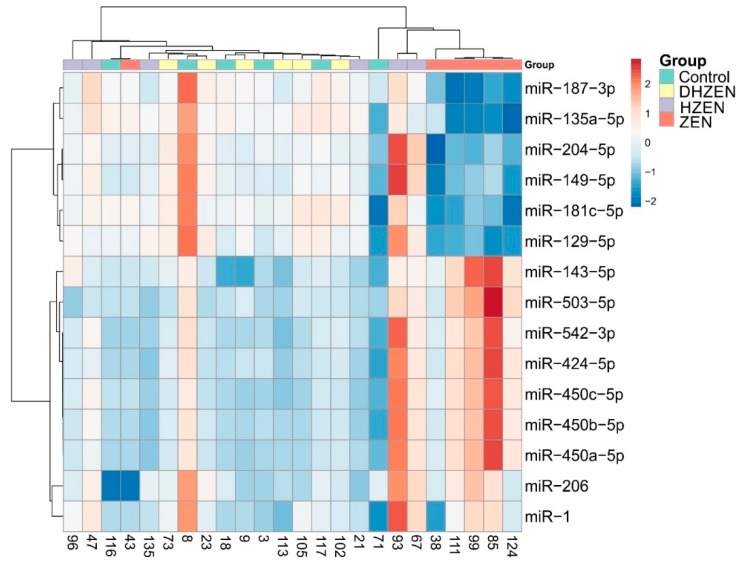
Hierarchical clustering analysis (average Pearson correlation coefficients) of microRNA expression in uterus of piglets exposed to uncontaminated feed (Control), ZEN (4.58 mg/kg), HZEN (4.84 mg/kg), or DHZEN (4.21 mg/kg) for 28 days (n = 6). The top row gives the treatment group, each column represents one pig. Each row represents one microRNA. Red rectangles indicate upregulation, blue rectangles indicate down regulation of the respective microRNA.

**Figure 5 toxins-11-00481-f005:**
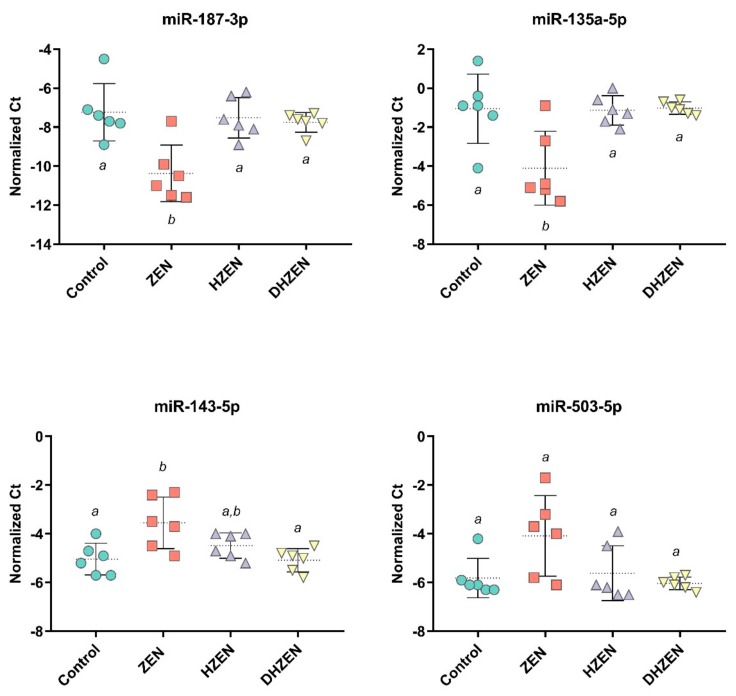
Normalized Ct-values of miR-187-3p, miR-135a-3p, miR-143-5p, and miR-503-5p in the uterus tissue of piglets exposed to uncontaminated feed (Control), ZEN, HZEN, or DHZEN (d28, n = 6). Individual symbols represent individual pigs. Significant differences between groups are indicated with dissimilar superscripts ^a,b^ (*p* < 0.05).

**Table 1 toxins-11-00481-t001:** Effect of dietary ZEN (4.58 mg/kg) and equimolar concentrations of HZEN (4.84 mg/kg) and DHZEN (4.21 mg/kg) on body weight, reproductive tract weight and vulva size of piglets (mean ± SD, n = 6) after 27 days of exposure. Significant differences between treatments are indicated with dissimilar superscripts ^a,b^ (*p* < 0.05).

Group	Body Weight (kg)	Vulva Size ^1^ (cm^2^)	Reproductive Tract Weight ^2^
Control	20.5 ± 2.8	1.47 ± 0.42^a^	51.8 ± 20.6^a^
ZEN	21.0 ± 2.2	4.84 ± 0.66^b^	353.4 ± 45.2^b^
HZEN	20.7 ± 2.5	1.70 ± 0.24^a^	59.8 ± 12.0^a^
DHZEN	20.5 ± 4.0	1.41 ± 0.27^a^	49.9 ± 11.1^a^

^1^ Calculated by multiplying vulva length (cm) with vulva with (cm), ^2^ Expressed in g per kg body weight × 100.

**Table 2 toxins-11-00481-t002:** Effect of dietary ZEN (4.58 mg/kg) and equimolar amounts of HZEN (4.84 mg/kg) and DHZEN (4.21 mg/kg) on gene expression in the uterus (d28, n = 6). Fold changes in treatment groups are expressed relative to the control group. Normalized Ct values (ΔCt-values) were used for statistical analysis. Significant differences between groups are indicated with dissimilar superscripts ^a,b^ (*p* < 0.05).

Gene Name	Protein Name	Relative Gene Expression Mean (and Range) Fold-Change Compared to Control			
		Control	ZEN	HZEN	DHZEN
**EBAG9**	Receptor-binding cancer antigen expressed on SiSo cells	1.00 ^a^ (0.61–1.65)	1.47 ^b^ (1.01–2.16)	1.04 ^a^ (0.67–1.61)	1.05 ^a^ (0.92–1.20)
**OVGP1**	Oviduct-specific glycoprotein or mucin-9	1.00 (0.54–1.85)	2.19 (1.03–4.70)	1.35 (0.68–2.67)	1.79 (1.07–3.00)
**IGFBP4**	Insulin-like growth factor-binding protein 4	1.00 (0.59–1.71)	0.57 (0.29–1.15)	1.16 (0.70–1.94)	1.23 (0.93–1.63)
**GJA1**	Gap junction alpha-1 protein or connexin 43	1.00 ^a^ (0.62–1.62)	1.93 ^b^ (1.48–2.52)	1.12 ^a,b^ (0.65–1.93)	1.26 ^a,b^ (0.98–1.62)
**GAPDH**	Glyceraldehyde 3phosphate dehydrogenase	1.00 ^a,b^ (0.46–2.15)	1.97 ^a^ (1.15–3.37)	1.08 ^a,b^ (0.48–2.43)	0.92 ^b^ (0.58–1.45)
**C3**	Complement component 3	1.00 (0.41–2.42)	0.78 (0.40–1.53)	0.86 (0.44–1.69)	1.40 (1.11–1.76)
**S100G**	S100 calciumbinding protein or calbindin D9K	1.00 ^a,b^ (0.13–7.72)	0.22 ^a^ (0.02–2.44)	2.48 ^a,b^ (1.68–3.66)	4.32 ^b^ (2.18–8.54)
**CLU**	Clusterin or apolipoprotein J	1.00 (0.53–1.90)	1.27 (0.65–2.50)	0.85 (0.53–1.35)	1.02 (0.73–1.44)
**ODC**	Ornithine decarboxylase	1.00 (0.55–1.82)	0.95 (0.71–1.28)	0.95 (0.62–1.45)	1.09 (0.89–1.33)
**ESR1**	Estrogen receptor alpha	1.00 (0.57–1.74)	1.11 (0.73–1.68)	1.18 (0.66–2.10)	1.24 (1.06–1.45)
**ESR2**	Estrogen receptor beta	1.00 (0.39–2.54)	1.22 (0.33–4.51)	0.40 (0.09–1.90)	1.62 (0.53–4.94)
**IL-1β**	Interleukin 1 beta	1.00 (0.26–3.90)	3.36 (0.47–24.12)	1.23 (0.43–3.52)	1.83 (1.12–3.01)
**IL-6**	Interleukin 6	1.00 (0.56–1.78)	0.89 (0.34–2.36)	1.40 (0.85–2.30)	1.90 (1.25–2.92)

**Table 3 toxins-11-00481-t003:** Effect of dietary ZEN (4.58 mg/kg) and equimolar amounts of HZEN (4.84 mg/kg) and DHZEN (4.21 mg/kg) on microRNA expression in the uterus (d 28, n = 6). Fold changes in treatment groups are expressed relative to the control group. Normalized Ct values (ΔCt-values) were used for statistical analysis. Significant differences between groups are indicated with dissimilar superscripts ^a,b^ (*p* < 0.05).

microRNA	Relative microRNA Expression Mean (and Range) Fold-Change Compared to Control			
	Control	ZEN	HZEN	DHZEN
**miR-1**	1.00 (0.47–2.11)	1.27 (0.72–2.22)	1.89 (0.63–5.66)	1.03 (0.67–1.60)
**miR-129-5p**	1.00 ^a,b^ (0.45–2.24)	0.42 ^a^ (0.18–0.99)	1.44 ^b^ (0.40–5.21)	1.19 ^b^ (0.78–1.82)
**miR-135a-5p**	1.00 ^a^ (0.32–3.11)	0.12 ^b^ (0.02–0.68)	0.94 ^a^ (0.21–4.27)	1.01 ^a^ (0.63–1.62)
**miR-143-5p**	1.00 ^a^ (0.47–2.14)	2.78 ^b^ (1.44–5.36)	1.45 ^a,b^ (0.38–5.48)	0.96 ^a^ (0.65–1.43)
**miR-149-5p**	1.00 ^a,b^ (0.46–2.18)	0.37 ^a^ (0.19–0.74)	1.77 ^b^ (0.59–5.31)	1.11 ^a,b^ (0.76–1.63)
**miR-181c-5p**	1.00 (0.40–2.52)	0.40 (0.17–0.96)	1.12 (0.30–4.24)	1.14 (0.70–1.87)
**miR-187-3p**	1.00 ^a^ (0.44–2.29)	0.11 ^b^ (0.03–0.48)	0.81 ^a^ (0.23–2.85)	0.70 ^a^ (0.42–1.16)
**miR-204-5p**	1.00 ^a^ (0.45–2.22)	0.29 ^b^ (0.13–0.66)	1.58 ^a^ (0.54–4.65)	0.99 ^a^ (0.66–1.47)
**miR-206**	1.00 (0.36–2.79)	1.34 (0.45–4.04)	1.88 (0.61–5.81)	0.88 (0.56–1.39)
**miR-424-5p**	1.00 (0.47–2.15)	3.69 (1.31–10.41)	1.86 (0.63–5.47)	0.95 (0.57–1.57)
**miR-450a-5p**	1.00 (0.50–1.98)	4.25 (1.45–12.42)	2.11 (0.72–6.21)	0.92 (0.60–1.42)
**miR-450b-5p**	1.00 (0.48–2.09)	3.63 (1.34–9.84)	2.00 (0.68–5.86)	0.94 (0.61–1.44)
**miR-450c-5p**	1.00 ^a,b^ (0.49–2.06)	4.52 ^a^ (1.60–12.75)	2.36 ^a,b^ (0.78–7.12)	0.84 ^b^ (0.54–1.30)
**miR-503-5p**	1.00 (0.53–1.89)	3.36 (1.29–8.74)	1.16 (0.38–3.51)	0.87 (0.49–1.54)
**miR-542-3p**	1.00 (0.47–2.12)	3.45 (1.20–9.89)	2.19 (0.72–6.63)	0.88 (0.54–1.42)

**Table 4 toxins-11-00481-t004:** Nucleotide sequence of primers for real-time qPCR.

Gene Name	Primer Sequence	Amplicon Size	Ensembl Access #
**Housekeeping Genes**			
HPRT1 ^1^	F (300 nM) GGACTTGAATCATGTTTGTG R (300 nM) CAGATGTTTCCAAACTCAAC	91 bp	ENSSSCG00000034896
RPL32 ^1^	F (300 nM) AGTTCATCCGGCACCAGTCA R (300 nM) GAACCTTCTCCGCACCCTGT	92 bp	ENSSSCG00000035811
RPL4 ^1^	F (300 nM) CAAGAGTAACTACAACCTTC R (300 nM) GAACTCTACGATGAATCTTC	122 bp	ENSSSCG00000004945
TBP ^1^	F (300 nM) AACAGTTCAGTAGTTATGAGCCAGA R (300 nM) AGATGTTCTCAAACGCTTCG	153 bp	ENSSSCG00000037372
**Genes with Identified ERE Motif**			
EBAG9	F (300 nM) GCACAGGTTTCTCTAGTAGGCT R (300 nM) TCCCTGTCTGCTATCTTCTGC	175 bp	ENSSSCG00000006024
OVGP1	F (300 nM) GGGTCGGCTATGATGATGACA R (300 nM) CCGGTGAAGGAGTTGAGCTA	198 bp	ENSSSCG00000006791
IGFBP4	F (300 nM) CATCCCCATCCCTAACTGCG R (300 nM) CTCACTCTCGGAAGCTGTCG	185 bp	ENSSSCG00000017472
GJA1	F (300 nM) TCTGAGTGCCTGAACTTGCT R (300 nM) CAGCGGTGGAATAGGCTTGA	154 bp	ENSSSCG00000004241
GAPDH ^1,2^	F (300 nM) AGGGGCTCTCCAGAACATCATCC R (300 nM) TCGCGTGCTCTTGCTGGGGTTGG	446 bp	ENSSSCG00000000694
C3 ^2^	F (300 nM) GGGCAGATCTTGAGTGTCCG R (300 nM) ATGCTGGATGAACTGAGCCC	179 bp	ENSSSCG00000013551
**Other Genes Associated With Estrogenic Response or Inflammation**			
S100G	F (300 nM) GGAGTTGAACTTGGACGTGC R (300 nM) CGCATCCCTTCCAGTCCTTA	184 bp	ENSSSCG00000012147
CLU	F (300 nM) CCATGACATGTTCCAGCCCT R (300 nM) TCTGAGAGGAATTGCTGGCC	239 bp	ENSSSCG00000009668
ODC	F (300 nM) CGGCGATTGGATGCTCTTTG R (300 nM) AAGTCGTGGTTCCGGATCTG	144 bp	ENSSSCG00000027121
ESR1	F (300 nM) CCTGGAGAATGAGCCGAGC R (300 nM) CTTCCCTTGTCACTGGTGCT	92 bp	ENSSSCG00000025777
ESR2	F (300 nM) TGCAGTGATTATGCGTCAGGA R (300 nM) CAGCTTTTACGCCGGTTCTT	149 bp	ENSSSCG00000005109
IL-1β	F (300 nM) CCATAGTACCTGAACCCGCC R (300 nM) GCTGGTGAGAGATTTGCAGC	165 bp	ENSSSCG00000039214
IL-6 ^1^	F (300 nM) GGCAAAAGGGAAAGAATCCAG R (300 nM) CGTTCTGTGACTGCAGCTTATCC	87 bp	ENSSSCG00000020970

^1^ The primers for these genes were already published in Nygard et al. [76], Grenier et al. [77], and Gessner et al. [78], and for the other genes, the primers were designed for the present study. ^2^ ERE motif was found in the human C3 and GAPDH sequences [48] but not in pig. In addition, we assessed GAPGH as both housekeeping gene and potential ERE gene. HPRT1, Hypoxanthine Phosphoribosyltransferase 1, RPL, Ribosomal Protein L, TBP, TATA-Box Binding Protein, GAPDH, Glyceraldehyde 3-phosphate dehydrogenase.

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
