# Peer review of "Biotransformation of the Mycotoxin Zearalenone to its Metabolites Hydrolyzed Zearalenone (HZEN) and Decarboxylated Hydrolyzed Zearalenone (DHZEN) Diminishes its Estrogenicity In Vitro and In Vivo"

_toxins, 2019, doi:10.3390/toxins11080481_

Round 1

Reviewer 1 Report

Use cell proliferation or survival instead WST-1 as keyword.

The figure 2 doesn’t show the effect of estradiol on cell proliferation. 

Authors assessed the effects of ZEN and etc. on MCF-7 cell lines but evaluated their in vivo effects  on uterus. Could you explain the discrepancy?

It is currently accepted that estrogens play a pivotal role also in males. Please read the following manuscript. doi: 10.3389/fonc.2018.00002.

What kind of enzyme the authors suppose to use to modify ZEN?

The possible use of enzymes to inactivate ZEN seems not available.

Reviewer 2 Report

Authors studied estrogenic activity of zearalenone and its metabolites using two in vitro assays and in vivo assay and demonstrated that the two metabolites of zearalenone have no estrogenic potency. Expression of microRNAs is hard to use for the detection of estrogenic activity at present.
For all the study is reasonably simple and easy to understand, therefore, I have no critical comments. Only one important future direction, based on these studies, need to be discussed in the last part of this manuscript. How to reduce estrogenicity (or how to metabolize zearalenone) in foods, feeds using degradation enzymes authors proposed. This is the objective of this study, therefore, authors may need to discuss this point. 

Minor editorial sug50, 56, 57, 60, 61, 63, gestions are as follows:

line 13: in vitro should be italic References should be checked: journal name should be appropriately abbreviated, ref no. 2, 5, 7, 8, 9, 11, 12, 17, 19, 21, 23, 24, 26, 30, 38, 41, 44, 45 Ref. 67, 68: Are these references correct? Ref. 32: Some authors name (Onose, Yamaguchi, Osada) should be revised.

Round 2

Reviewer 1 Report

The current form of the manuscript is suitable for publication in Toxins.